# Patterns of Relapse in Small Cell Lung Cancer: Competing Risks of Thoracic versus CNS Relapse

Peter M. Ellis [1,2,*], Anand Swaminath [1,3] and Gregory R. Pond [2]

1   Department of Oncology, McMaster University, Main St. W., Hamilton, ON L8S 4L8, Canada; swaminath@hhsc.ca
2   Juravinski Cancer Centre, Division of Medical Oncology, 699 Concession St., Hamilton, ON L8V 5C2, Canada; gpond@mcmaster.ca
3   Juravinski Cancer Centre, Division of Radiation Oncology, 699 Concession St., Hamilton, ON L8V 5C2, Canada
*   Correspondence: ellisp@hhsc.ca; Tel.: +1-905-387-9495; Fax: +1-905-575-6326

**Abstract:** Introduction: Treatment algorithms for small cell lung cancer (SCLC) are determined largely by the Veterans Affairs Lung Cancer Staging Group (VALCSG) staging (limited (LS) versus extensive (ES) stage). Relapse occurs frequently; however, patterns of relapse, in particular the competing risk of thoracic and central nervous system relapse, are not well described. This study describes patterns of relapse in SCLC patients treated at a large tertiary institution in Ontario, Canada. Materials and Methods: A retrospective cohort of SCLC patients treated at the Juravinski Cancer Centre was reviewed. Data were abstracted from the medical record on demographic, disease, treatment and outcome variables. The primary outcome was a description of the patterns of relapse stratified by disease stage. Multivariate analysis was performed to identify prognostic variables for thoracic and CNS relapse. Results: Two hundred and twenty nine patients were treated during the study period (LS—83, ES—146). Relapse occurred in the majority of patients (isolated thoracic—28%, isolated CNS—9%, extrathoracic—9%, thoracic/extrathoracic—14%, systemic and CNS—13%). The median OS was consistent with published data (LS—21.8 months, ES—8.9 months). ES disease and elevated LDH were prognostic for increased thoracic relapse, whereas poor PS and older age were prognostic for lower central nervous system (CNS) relapse. Discussion: Thoracic relapse and CNS relapse represent competing risks for patients with SCLC. Decisions about incorporating thoracic or CNS radiation are complex. More research is needed to incorporate performance status and LDH into treatment algorithms.

**Keywords:** small cell lung cancer; recurrent disease; health outcomes; competing risk





## 1. Introduction

Small cell lung cancer (SCLC) accounts for only 12–15% of lung cancer cases [1,2]. Nevertheless, this still represents a significant health burden, with over 4000 cases annually across Canada [3]. Historically, a treatment-based staging system developed by the Veterans Affairs Lung Cancer Study Group (VALCSG) was used in SCLC [4], with disease confined to one hemithorax that could be encompassed in a single radiation field classified as limited stage (LS) disease, and everything else was classified as extensive stage (ES) disease. In recent years, however, there is a move to anatomic staging, using the TNM system [5].

The primary treatment of SCLC is platinum-based chemotherapy, with thoracic radiation routinely added to the management of patients with LS SCLC [6]. While there is a high likelihood of initial response to therapy, the risk of recurrence is high. Patients with SCLC are also at high risk for the development of brain metastases, and a meta-analysis of randomized trials supports the use of prophylactic cranial irradiation (PCI) in LS SCLC patients achieving a complete response to their initial chemoradiation therapy [7]. There has been an interest in expanding the role for radiation in ES SCLC in the last decade. However, randomized trials have demonstrated conflicting results with respect to overall survival

(OS) from PCI in patients responding to initial chemotherapy [8] and a non-definitive suggestion of improved OS from thoracic radiation [9]. The gains in OS from these treatments are modest at best and the selection of patients most likely to benefit remains challenging.

Decisions regarding the use of thoracic radiation and PCI are influenced by the likelihood of relapse, in addition to published data. However, the body of literature describing patterns of relapse, in particular the competing risks of central nervous system (CNS) and systemic recurrence, is small. Giuliani et al. reported on a consecutive series of 253 patients with LS SCLC seen over 12 years [10]. The first site of recurrence was locoregional in 34 people, distant in 80 and locoregional plus distant in 31. Brain metastases were detected in 57 patients and CNS recurrence was higher in patients who did not receive PCI (43% vs. 23%). A retrospective review of 28 Japanese patients undergoing surgical resection for SCLC reported recurrence in 10 of 29 patients [11]. Two patients had local recurrence alone, one had local and distant and seven had distant recurrence (four CNS). Arriagada et al. [12] reported on the competing risk of local recurrence, distant recurrence, or death without recurrence. At two years, the cumulative incidence of local recurrence was 33%, distant recurrence 25% and local plus distant recurrence 10%. However, more data are needed evaluating the use of thoracic radiation and PCI, together with patterns of recurrence, particularly in ES SCLC, as well as the competing risk of systemic versus CNS recurrence. Understanding the issue of competing risk may help improve the selection of patients for PCI and thoracic radiation. The current study examined patterns of recurrence among SCLC patients treated at a large tertiary care institution in Ontario, Canada.

## 2. Materials and Methods

This study was a retrospective cohort study of consecutive new patients with SCLC receiving systemic therapy at a tertiary cancer centre in Ontario, Canada between January 2011 and December 2014. Follow up continued until June 2016. The minimum follow up period was 18 months. The centre is a comprehensive tertiary centre for cancer care and research in Ontario, Canada. Patients with SCLC were identified from a search of the institutional electronic database. Patients with mixed histology (small cell and non small cell) and patients who were only seen for radiation therapy were excluded from the current study. Patients seen during this period were routinely staged according to the VALCSG staging system.

Data were extracted from the medical record, along with hospital laboratory and radiology databases including: patient demographics such as age and sex, smoking status, performance status (where recorded), staging information, systemic treatment details including physician assessed response to therapy, details of any radiation therapy, timing and patterns of relapse and survival information. Outcome of treatment was extracted from the patients' medical record and the patients' date of last contact and vital status recorded. The patients' family doctor was contacted to determine the date of death, or date of last contact, for patients who had not been seen within the last three months. The study was approved by the local Research Ethics Board.

The primary outcome of this study was a description of the patterns of relapse. This study was reflective of clinical practice. Most patients were followed with CT imaging of the chest +/− abdomen. The schedule of follow up was determined by individual physicians and not standardized. Brain imaging was typically only performed to investigate new onset of CNS symptoms. Various patterns of recurrence were defined in advance including: thoracic only, distant relapse only, central nervous system (CNS) only, thoracic and distant, or combined systemic and CNS recurrence. Secondary outcomes included overall response rate (ORR) as defined by the treating physician, progression free survival (PFS), overall survival (OS), as well as the cumulative incidence of thoracic and CNS sites of metastatic disease. Outcome data were assessed for the entire population, as well as stratified according to stage (LS or ES). PFS was defined as time from diagnosis to documented relapse or death, recognizing that response assessments were not conducted

on a standardized schedule. OS was defined as the time from diagnosis to death from any cause.

Data were analyzed using SPSS V25. Data summaries, including patterns of relapse, were presented for the entire population, as well as the LS and ES patient groups. All variables were summarized using frequency tables. Patients known to be alive at the time of last follow up were censored and PFS and OS were calculated according to the Kaplan–Meier method. Cumulative incidence curves were generated to calculate the risk of thoracic and CNS relapse at 12 and 24 months. Cox regression analysis was used to evaluate variables that were prognostic for OS. Separate multivariable models were constructed to identify prognostic variables for the risk of CNS and thoracic relapse, each outcome using a full model, which included all clinically available patient characteristics, including patient gender, performance status, use of thoracic radiation, PCI, LDH, age, stage, and weight loss prior to baseline. The chi-square test was used to compare between types of relapse based on treatment received. Statistical significance was defined at the alpha = 0.05 level, and all comparisons and confidence intervals were two-sided.

### 3. Results

There were 275 patients with a new diagnosis of SCLC seen during the study period. Forty six (16.7%) patients received no systemic treatment and are not included in this report. The baseline demographic characteristics of the 229 treated patients are summarized in Table 1. Approximately one third had LS disease at diagnosis and two thirds ES disease. The median age was 66 years, with approximately half the cohort female. Almost all patients were current or former smokers. Poor performance status (ECOG 3–4) was observed in 13% of the treated population, with 20% having an elevated lactate dehydrogenase (LDH) and 33% with weight loss greater than 5%. Patients with ES SCLC were more likely to be current smokers, have an elevated LDH, have poorer performance status and have lost more than 5% body weight.

**Table 1.** Patient characteristics.

| Variable | | Limited Stage | Extensive Stage | Total |
|---|---|---|---|---|
| | | **83** | **146** | **229** |
| Sex | Male | 37 (44.6%) | 78 (53.4%) | 115 (50.2%) |
| Age (yrs) | Median (std) | 68 (9.0) | 65 (9.4) | 66.0 (9.3) |
| Smoking Status | never smoked | 0 | 3 (2%) | 3 (1.3%) |
| | current | 44 (53%) | 89 (61%) | 133 (58.1%) |
| | former | 39 (47%) | 54 (37%) | 93 (40.6%) |
| LDH | Normal | 37 (44.6%) | 48 (32.9%) | 85 (37.1%) |
| | Elevated | 26 (31.3%) | 72 (49.3%) | 98 (42.8%) |
| | Missing | 20 (24.1%) | 26 (17.8%) | 46 (20.1%) |
| ECOG PS | 0–1 | 66 (79.5%) | 67 (58.1%) | 133 (58.1) |
| | 2 | 10 (12%) | 66 (28.8%) | 66 (28.8) |
| | 3–4 | 7 (8.4%) | 30 (13.1%) | 30 (13.1) |
| Weight Loss | WL < 5% | 65 (78.3%) | 89 (61%) | 154 (67.2) |
| | WL > 5% | 18 (21.7%) | 57 (39%) | 75 (32.8) |
| Sites of Metastatic Disease | Pleural effusion | - | 34 (23.3%) | 34 (14.8%) |
| | Brain | - | 36 (24.7%) | 35 (15.7%) |
| | Liver | - | 70 (47.9%) | 70 (30.6%) |
| | Adrenal | - | 28 (19.2%) | 28 (12.2%) |
| | Bone | - | 50 (34.2%) | 50 (21.8%) |

LDH = lactate dehydrogenase.

### 3.1. Treatment and Survival Outcomes

Treatment details of the cohort are summarized in Table 2. Almost all patients were treated with a platinum agent plus etoposide. Most patients received between four and six cycles of chemotherapy, although more patients with ES SCLC received fewer than four cycles of treatment. The majority of patients with LS SCLC received thoracic radiation (87%), but thoracic radiation was given to only 27% of patients with ES disease. Prophylactic cranial irradiation (PCI) was received by 59% of LS patients and only 21% of patients with ES disease. The dose of PCI was 24 Gy in 10 fractions. Reasons for not giving PCI are summarized in Table 2. Patients with established brain metastases were typically treated with either 30 Gy in 10 fractions, or 20 Gy in 5 fractions. The median OS for patients with LS SCLC was 21.8 months versus 8.9 months for patients with ES SCLC (Figure 1). A Cox regression analysis was undertaken to determine prognostic factors for OS (Table 3). Female gender, better PS, receipt of thoracic radiation, receipt of PCI and normal LDH were all prognostic for improved OS.

**Table 2.** Summary of treatments received.

| Variable | | Limited Stage | Extensive Stage | Total |
|---|---|---|---|---|
| | | **83** | **146** | **229** |
| Chemotherapy | cisplatin + etoposide | 49 (59%) | 72 (49.3%) | 121 (52.8%) |
| | carboplatin + etoposide | 33 (39.8%) | 66 (45.2%) | 99 (43.2%) |
| | oral etoposide | 1 (1.2%) | 8 (5.5%) | 9 (3.9%) |
| Number of Cycles | <4 | 12 (14.4%) | 44 (30.1%) | 56 (24.4%) |
| | 4 | 34 (41%) | 19 (13.0%) | 53 (23.1%) |
| | 5–6 | 37 (44.6%) | 83 (56.9%) | 120 (52.4%) |
| Response | progressive disease | 4 (4.8%) | 32 (21.9%) | 36 (15.7%) |
| | stable disease | 7 (8.4%) | 29 (19.9%) | 36 (15.7%) |
| | partial response | 59 (71.1%) | 81 (55.5%) | 140 (61.1%) |
| | complete response | 13 (15.7%) | 4 (2.7%) | 17 (7.4%) |
| Thoracic Radiation | none | 10 (12.7%) | 107 (73.3%) | 117 (51.1%) |
| | pre chemotherapy | - | 7 (4.8%) | 7 (3.1%) |
| | C 1 or 2 chemo | 50 (60.3%) | 7 (4.8%) | 57 (24.9%) |
| | C3 or higher | 19 (22.9%) | 4 (2.8%) | 23 (10%) |
| | post chemotherapy | 4 (4.8%) | 21 (14.4%) | 25 (10.9%) |
| Radiation Dose | no radiation | 10 (12.7%) | 107 (73.3%) | 117 (51.1%) |
| | 50 Gy/25 fraction | 49 (59%) | 4 (2.7%) | 53 (23.1%) |
| | 45 Gy/30 fractions BID | 18 (21.7%) | 4 (2.7%) | 22 (9.6%) |
| | 40 Gy/15 fractions | 6 (7.2%) | 6 (4.1%) | 12 (5.2%) |
| | palliative radiation only | - | 25 (17.1%) | 25 (10.9%) |
| PCI | no PCI, patient choice | 20 (24.1%) | 21 (14.4%) | 41 (17.9%) |
| | no PCI, physician advise | 4 (4.8%) | 20 (13.7%) | 24 (10.5%) |
| | brain mets, no PCI | 0 | 34 (23.3%) | 34 (14.8%) |
| | no PCI, other health issues | 6 (7.2%) | 16 (11%) | 22 (9.6%) |
| | no PCI, disease progression | 4 (4.8%) | 24 (16.4%) | 28 (12.2%) |
| | Yes | 49 (59%) | 31 (21.2%) | 80 (34.9%) |

C = cycle, GY = Gray, BID = twice daily, PCI = prophylactic cranial irradiation.

**Table 3.** Results of multivariate analyses.

| Variable | OS * | | Thoracic Relapse | | CNS Relapse | |
|---|---|---|---|---|---|---|
| | HR (95% CI) | *p* Value | HR (95% CI) | *p* Value | HR (95% CI) | *p* Value |
| Gender | | | | | | |
| Male | 1 | 0.001 | 1 | 0.74 | 1 | 0.12 |
| Female | 0.6 (0.45–0.81) | | 1.05 (0.61–1.82) | | 1.72 (0.87–3.37) | |
| PS | | | | | | |
| 0 | 1 | | 1 | | 1 | |
| 1 | 1.58 (0.93–2.7) | 0.037 | 1.22 (0.48–3.31) | 0.64 | 0.19 (0.07–0.51) | |
| 2 | 1.76 (0.99–3.1) | | 0.72 (0.25–2.04) | | 0.16 (0.05–0.47) | 0.01 |
| 3 | 2.84 (1.47–5.48) | | 1.17 (0.34–4.02) | | 0.17 (0.04–0.66) | |
| 4 | 2.04 (0.77–5.38) | | 1.55 (0.20–11.7) | | 0 (0.0–NR) | |
| Thoracic radiation | 0.46 (0.34–0.64) | <0.001 | 0.86 (0.44–1.68) | 0.68 | 1.16 (0.50 -2.71) | 0.61 |
| PCI | 0.45 (0.32–0.64) | <0.001 | 1.43 (0.76–2.71) | 0.27 | 0.68 (0.31–1.5) | 0.2 |
| LDH | | | | | | |
| <220 U/L | 1 | <0.001 | 1 | 0.075 | 1 | |
| >220 U/L | 2.25 (1.58–3.19) | | 1.19 (0.65–2.17) | | 1.38 (0.62–3.1) | 0.58 |
| unknown | 1.2 (0.78–1.84) | | 0.51 (0.24–1.07) | | 0.86 (0.31–2.37) | |
| Age | 1 | 0.95 | 0.99 (0.97–1.03) | 0.83 | 0.94 (0.90–0.98) | 0.002 |
| Stage | | | | | | |
| LS | 1 | 0.37 | 1 | 0.007 | 1 | 0.17 |
| ES | 1.2 (0.78–1.84) | | 2.18 (1.24–3.83) | | 1.81 (0.82–3.95) | |
| Weight loss | | | | | | |
| <5% | 1 | 0.29 | 1 | 0.79 | 1 | 0.43 |
| ≥5% | 1.18 (0.87–1.62) | | 0.97 (0.53–1.78) | | 0.68 (0.31–1.48) | |

* Cox regression analysis, PS = Performance Status, PCI = Prophylactic Cranial Irradiation, LDH = lactate dehydrogenase, LS = limited stage, ES = extensive stage.

### 3.2. Patterns of Relapse

Relapses were documented in 64% of patients with LS disease and 78% of patients with ES disease (Table 4). The patterns of relapse are described in Table 4. Isolated thoracic relapse (28%) and CNS relapse (9%) occur commonly. Thoracic recurrence was observed in 40% of patients with LS disease compared with 60% of patients with ES disease. Fewer thoracic recurrences were observed in patients who received thoracic radiation (46% versus 59%). For patients with LS SCLC, thoracic recurrence was observed in 27 (37%) of patients who received thoracic radiation and 6 (60%) who did not. Most thoracic recurrences were confined to the chest, rather than chest plus other sites (29% versus 11%). Among patients with ES SCLC, thoracic recurrence was observed in 25 (64%) patients who received thoracic radiation and 63 (59%) who did not. A similar number of thoracic recurrences

were confined to the chest, versus chest plus other sites (27% vs. 33%). The cumulative incidence of thoracic relapse at 12 and 24 months was 31% and 47.2% for LS SCLC and 60.5% and 68.1% for ES SCLC (Figure 2a). Receipt of PCI was not associated with the risk of thoracic relapse (*p* = 0.84).

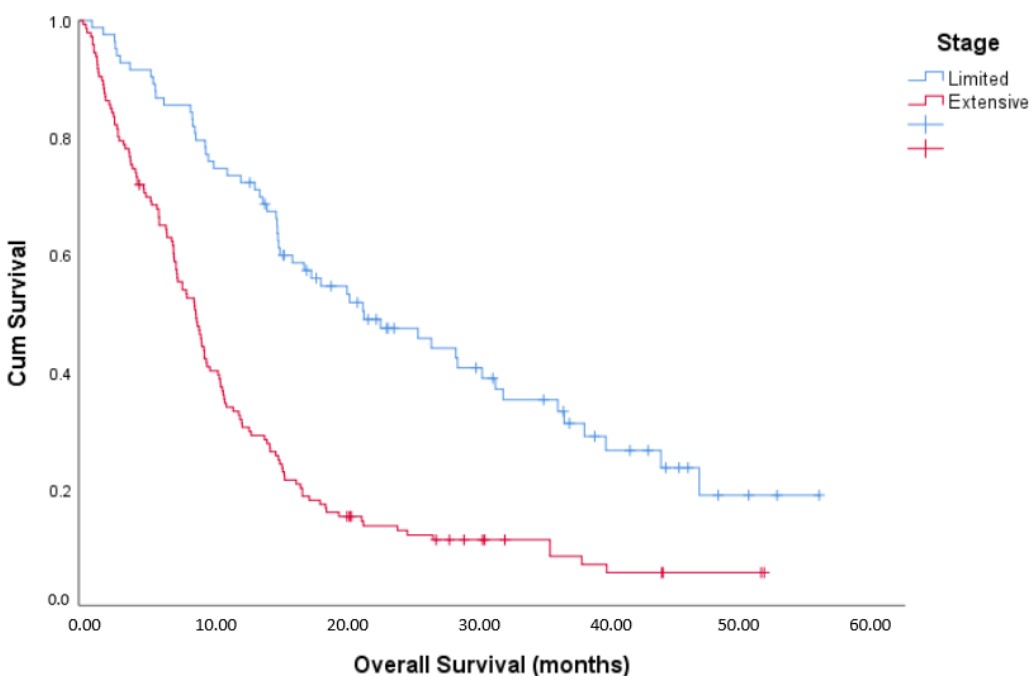

**Figure 1.** Kaplan–Meier overall survival curves.

**Table 4.** Patterns of relapse and outcomes of treatment.

| Variable | | Limited Stage | Extensive Stage | Total |
|---|---|---|---|---|
| | | **83** | **146** | **229** |
| Overall Survival | N (%) deaths | 55 (66.3%) | 132 (90.4%) | 187 (81.7%) |
| | Median OS | 21.8 m | 8.9 m | 11.1 m |
| | 1 year OS | 73.50% | 33.20% | |
| | 2 year OS | 47.40% | 13.60% | |
| Progression Free Survival | N (%) of relapses | 53 (63.9%) | 114 (78.1%) | 167 (72.9%) |
| | Median PFS | 14.3 m | 7.5 m | 9.2 m |
| Pattern of Relapse | No relapse | 30 (36.1%) | 32 (21.9%) | 62 (27.1%) |
| | Thoracic relapse | 24 (28.9%) | 40 (27.4%) | 64 (27.9%) |
| | Extra-thoracic relapse | 8 (9.6%) | 12 (8.2%) | 20 (8.7%) |
| | CNS relapse | 9 (10.8%) | 11 (7.5%) | 20 (8.7%) |
| | Combined systemic | 5 (6%) | 28 (19.2%) | 33 (14.4%) |
| | Combined systemic + CNS | 7 (8.4%) | 23 (15.8%) | 30 (13.1%) |
| Cumulative incidence of thoracic relapse | N (%) of thoracic relapse | 33 (39.8%) | 88 (60.3%) | |
| | 12 month incidence | 31.00% | 60.50% | 121 (52.8%) |
| | 24 month incidence | 47.20% | 68.10% | |
| Cumulative incidence of CNS relapse | N (%) of CNS relapse | 16 (19.3%) | 34 (23.3%) | |
| | 12 month incidence | 13.10% | 16.00% | 50 (21.8%) |
| | 24 month incidence | 19.10% | 19.50% | |

CNS = central nervous system, Combined systemic = thoracic plus extrathoracic.

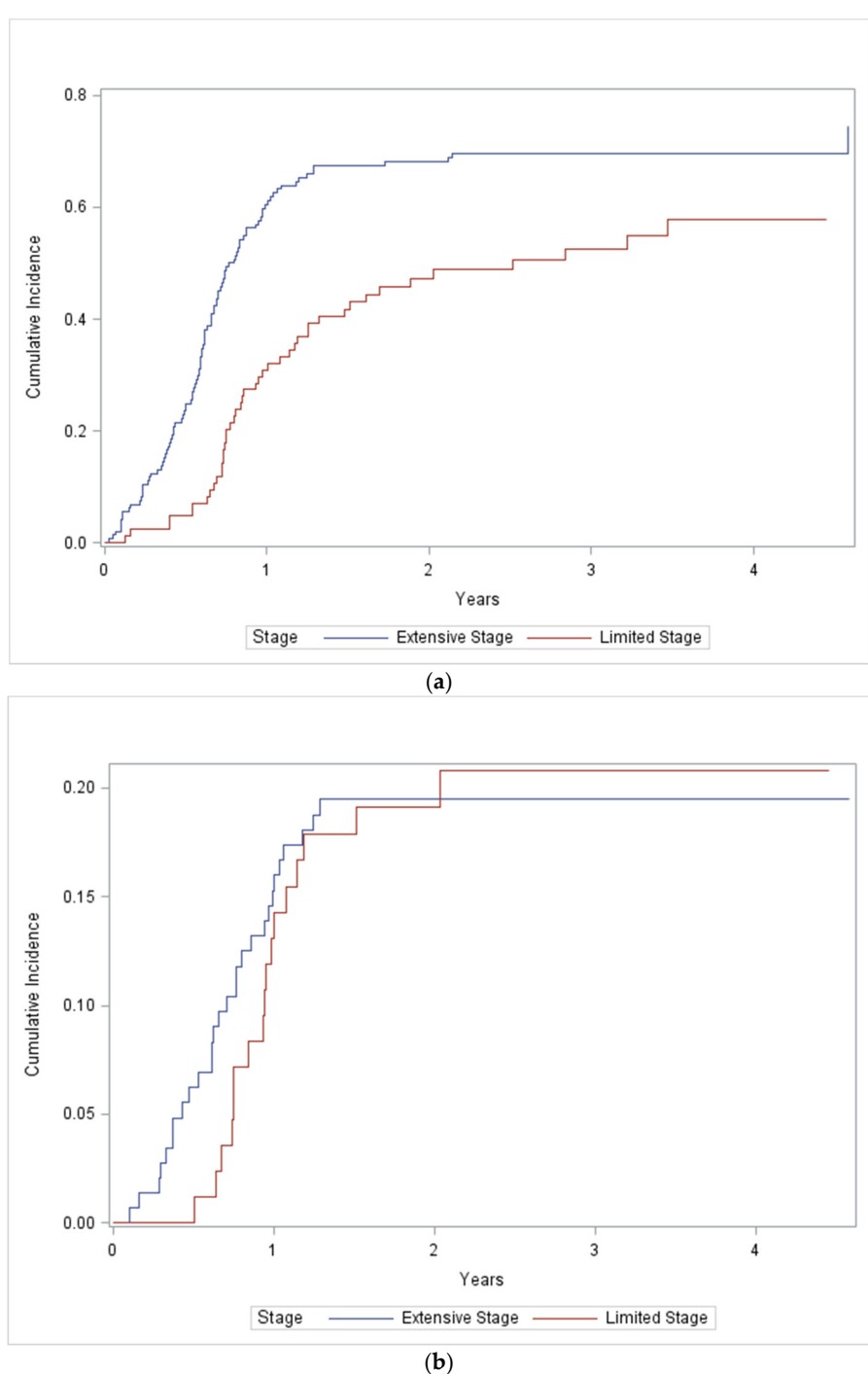

**Figure 2.** Cumulative incidence of thoracic (**a**) and CNS (**b**) recurrence.

CNS recurrence occurred less commonly than thoracic recurrence. Overall, 22% of patients experienced CNS recurrence, with a similar incidence in both LS and ES SCLC (19% vs. 23%). Similar rates of CNS recurrence in the entire cohort were documented in patients who did or did not receive PCI (20% vs. 23%). For patients with LS SCLC, CNS recurrence occurred in nine (18%) patients who did and seven (21%) patients who did not receive PCI. There were similar numbers of CNS only and CNS plus systemic recurrences (11% vs. 8%). For patients with ES SCLC, CNS recurrence occurred in 7 (23%) patients who received PCI and 27 (24%) patients who did not receive PCI. Fewer patients with ES SCLC had CNS as the only site recurrence (8% vs. 16%). The cumulative incidence of CNS relapse at 12 and 24 months was 13% and 19% for LS SCLC and 16% and 20% for ES SCLC

(Figure 2b). Receipt of thoracic radiation was not associated with the risk of CNS relapse (*p* = 0.62).

Multivariate analyses were undertaken to identify factors prognostic for thoracic recurrence and CNS recurrence. Gender, PS, age, stage, LDH, weight loss and receipt of thoracic radiation and PCI were included in the model (Table 3). ES disease (HR 2.18, 95% CI 1.24–3.83) and elevated LDH (HR 1.19, 95% CI 0.65–2.17) were prognostic for an increased risk of thoracic relapse. Worse PS and older age were both prognostic for a reduced risk of CNS relapse. Receipt of thoracic radiation or PCI was not prognostic for either thoracic or CNS recurrence after adjustment for other prognostic variables.

## 4. Discussion

This study provides valuable insight into the patterns of recurrence of SCLC. The patient demographics such as sex, age and stage distribution are representative of population data for SCLC patients in Ontario [2]. Many patients are elderly, with the median age at diagnosis approaching 70. Thirty to forty percent of patients present with adverse prognostic factors such as elevated LDH (43%), poor performance status (42%) and weight loss greater than 5% (33%), contributing to the aggressive nature of this disease and relatively poor outlook, particularly for ES SCLC. Among patients with ES SCLC, nearly 25% have brain metastases at the time of diagnosis. Patients often present with symptomatic and advanced disease and nearly one in six patients presented too unwell for systemic treatment and were treated with supportive management alone. Reassuringly, the outcome data for this cohort of treated patients are similar to previously published data, with median OS for LS SCLC 21.8 months and ES SCLC 8.9 months. Factors prognostic for overall survival include female gender, better performance status, normal LDH, along with receipt of either thoracic radiation or PCI.

Treatment patterns for this cohort of patients reflect guideline-recommended therapies [13]. The majority of patients with LS SCLC received chemotherapy and thoracic radiation, with many receiving PCI as well. Most patients with ES SCLC received treatment with systemic therapy alone. The low uptake of both thoracic radiation and PCI likely reflects marginal benefits from these interventions [8,9] in the context of limited survival. This dataset demonstrates that relapse occurs in the majority of patients with SCLC. Not surprisingly, more patients with ES SCLC have relapse of their disease. Interestingly though, there are considerable similarities in the frequency of isolated thoracic, extrathoracic and CNS relapses between LS and ES SCLC, although patients with ES SCLC are more likely to have combined systemic and CNS relapses.

Thoracic relapse occurred commonly in this cohort of patients and represents the most common site of relapse for SCLC patients. Turrissi reported local failure in 36% of LS SCLC patients treated with twice daily thoracic radiation and 52% of patients treated with daily radiation [14]. In this cohort, both daily and twice daily radiation schedules were used. The risk of thoracic recurrence (46%) was similar to that reported by Turrissi. In contrast, a single institution report by Guilani et al. reported locoregional failure in only 15% of LS SCLC patients [10]. This may reflect issues of ascertainment bias in retrospective reviews. Not surprisingly, thoracic relapse was less common in patients who received thoracic radiation. Interestingly, thoracic relapses in LS SCLC were much more likely to be locoregional rather than locoregional plus distant. This differed in ES SCLC, where there were similar numbers of locoregional only and locoregional plus distant relapses. The timing of thoracic relapse also differed between LS and ES SCLC. In LS SCLC, the risk of thoracic failure increased progressively over two years (31% at 1 year, 47% at 2 years), in contrast to ES SCLC where the risk of thoracic relapse peaked by the end of the first year.

The pattern of CNS recurrence in this cohort of patients differs from other published data. Randomized trials of prophylactic cranial irradiation have consistently demonstrated greater than a 50% reduction in the risk of brain metastases in both LS SCLC [7] and ES SCLC [8]. Receipt of PCI did not appear to influence the risk of symptomatic brain metastases in this cohort of patients. The incidence of brain metastases was similar in both

LS and ES SCLC and was similar in patients who did and did not receive PCI. Similar to thoracic radiation, the risk of brain metastases in LS SCLC increased progressively over two years, whereas the risk plateaued by one year in ES SCLC. The cumulative incidence of CNS relapse was only 22% though. One in four patients with ES SCLC had brain metastases at diagnosis. Patients in this cohort routinely underwent brain imaging at diagnosis, frequently with MRI. It is possible that more patients were diagnosed with brain metastases at diagnosis and this influenced the subsequent risk of CNS relapse. However, this finding requires further validation. It is also possible that patients outside of clinical trials have worse prognosis and many of these patients may have died prior to observing a benefit from PCI.

Thoracic and CNS relapse represent competing risks. The risk of thoracic recurrence is higher than CNS recurrence and both thoracic and CNS recurrences occur earlier in ES SCLC than LS SCLC. In multivariate analysis, extensive stage and elevated LDH are the only factors that appear prognostic for thoracic relapse. These findings suggest that disease-related factors are more important prognostic factors for thoracic recurrence than treatment variables. However, higher performance status and older age appear prognostic for reduced risk of CNS relapse. This may reflect that patients with a shorter expected survival die before they have a chance to develop CNS recurrence, reinforcing the point above that patients in routine practice have worse prognosis than patients in clinical trials.

This retrospective cohort analysis does not provide answers to the effectiveness of thoracic radiation, or PCI. That is best determined using data from randomized clinical trials. The findings, however, may help clinicians to conceptualize the issue of competing risk and individualize radiation treatment decisions in SCLC. Thoracic recurrence is two or three times more likely to occur than CNS recurrence. This suggests that decisions about the implementation of thoracic radiation may be more important to consider than PCI, particularly in ES SCLC. The findings suggest that LDH should be routinely estimated at baseline to identify ES SCLC patients at increased risk for thoracic recurrence. Early thoracic recurrence may mitigate any benefit from PCI, and therefore elevated LDH at diagnosis may have greater utility in decision making around PCI in ES SCLC. LDH was not examined in trials of either thoracic radiation or PCI in ES SCLC [8,9]. Similarly, poor performance and increasing age are associated with less risk of CNS recurrence and may help in selecting patients less likely to benefit from PCI. However, these observations require prospective validation.

There are limitations to these data. The data were collected retrospectively and some data were missing. The decisions to treat patients with thoracic radiation or PCI were clinician-based and not randomized. Disease burden and prognosis are already likely to have been factored into the radiation treatment decisions. However, many of these patients would not have been eligible for clinical trials evaluating thoracic radiation and PCI and would require extrapolation of the trial data. Additionally, these data do not include patients treated with immunotherapy. Recent data from the ImPower133 and Caspian trials demonstrate modest improvements in overall survival from the addition of atezolizumab or durvalumab to standard platinum and etoposide chemotherapy [15,16]. It is unclear, though, if the addition of an immune checkpoint inhibitor will alter the pattern of recurrence in these patients. Nevertheless, these findings provide some guidance in the extrapolation of clinical trial data to real world populations of patients with SCLC. These data demonstrate relatively low rates of CNS recurrence. CNS imaging was not standardized and therefore may have underestimated the true incidence of CNS recurrence.

## 5. Conclusions

In conclusion, these data demonstrate the complexity of decision making in regard to the addition of radiation to systemic therapy in SCLC. Patients treated with thoracic radiation and PCI have longer survival, and yet it is disease-related variables that appear to be prognostic for thoracic or CNS recurrence. Further research is required to understand

how to more effectively incorporate baseline prognostic variables such as performance status, LDH and age into treatment algorithms for SCLC.

**Author Contributions:** P.M.E. and A.S. contributed to study concepts and design. P.M.E. performed the literature search. P.M.E. and G.R.P. contributed to statistical analysis. Manuscript preparation was undertaken by P.M.E. All authors contributed to manuscript editing and approved the final version. All authors have read and agreed to the published version of the manuscript.

**Funding:** This research received no external funding.

**Institutional Review Board Statement:** This study was approved by the Hamilton Integrated Research Ethics Board.

**Informed Consent Statement:** NA as retrospective chart review.

**Data Availability Statement:** The data are not publicly available in accordance with the Research Ethics Board conditions of approval.

**Acknowledgments:** Acknowledgements to Abdulaziz Al Farsi, who collected and entered the data from the patient medical records.

**Conflicts of Interest:** Ellis has received honoraria for advisory boards from AstraZeneca, Pfizer and Takeda, plus honoraria for speaking from AstraZeneca, Eli Lilly, Pfizer and Bristol Meyers Squibb. Swaminath has received honoraria for advisory boards from AstraZeneca and Bristol Meyers Squibb. Pond has received honoraria for advisory boards from AstraZeneca and Merck and honoraria from Takeda for DSMB activities.

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
