# Peer review of "Patterns of Relapse in Small Cell Lung Cancer: Competing Risks of Thoracic versus CNS Relapse"

_curroncol, doi:10.3390/curroncol28040243_

Round 1

Reviewer 1 Report

the dose per fraction of WBRT would be good to add if it is available

Author Response

Please see the attachment covering all comments

Reviewer 2 Report

This report investigated the patterns of relapse, including the competing risk of thoracic and central nervous system relapse of SCLC patients in a single center. There are some issues needed to be clarified.

Major concern:

  1. The authors investigate the pattern of relapse in patients with SCLC. However, some detail should be provided. For example, the duration of follow up in this cohort? How the disease status was monitored? Did all the patients receive computed tomography (CT) of chest in a regular interval? How the CNS status was monitored? Did the patients receive brain CT/or brain MRI regularly?
  2. Following the first question, the relapse rate in ES disease was 78% in this study, which is lower than expected. If the patient expired before radiological documented relapse, the patients was classified as stable disease in this cohort. However, the actual relapse rate may be underestimated due to irregular follow up.
  3. The CNS relapse rate was 19.3% in LS-SCLC and 23.3% ES-SCLC, respectively. Again, the actual CNS relapse rate may be underestimated due to irregular follow up. Asymptomatic brain metastatic lesions may be detected by routine brain imaging checkup.
  4. The authors reported that poor ECOG PS and older age were associated with lower risk of CNS relapse, and maybe get less benefit from PCI. This finding may be partly explained by selection bias, and need be clarified due to these patients may be expired before documented CNS relapse. PCI is not suitable for patients with poor performance status and old age due to the concern of neurocognitive decline, but not the lower CNS relapse rate in these group. (discussion, page 7, line 262-264)

Minor concern:

  1. The numbers of relapse rate should match Table 3.
  2. There are many abbreviations in the abstract and some unnecessary words. Please describe full name when first use and amend the abstract.   
  3. Page 2, line 77. “the first site of recurrence was locoregional in 34, …” Please provide unit.
  4. Table 1: site of metastatic disease. Did patients with pericardial effusion was grouped in “Pleural effusion”? How about patients with lung to-lung metastasis?
  5. Table 1: age. Please provide the range of age.
  6. Table 2. Too many abbreviations, such as C1 or C2 chemo; fractions BID…. Please consider amened Table 2.
  7. Page 6, line 239. It is not appropriate to end a paragraph with question mark in an academic paper. Please consider re-white the sentence.
  8. Page 7, line 273. improvements in overall “survival”
  9. Table 4. Please provide LDH unit.

Author Response

(The authors gave the same response as above.)

Reviewer 3 Report

The authors report a series of 229 SCLC treated with chemotherapy +/-RT or PCI followed up to detect the different incidence of relapses based on original extension of disease and evaluating possible prognosticators for thoracic or CNS recurrence. The results are as expected and already reported in literature, such as the prognosticators. The paper appears basic in the aims, no clear additional practical and clinical message is reported and the methods are also not clear.

In particular it is unclear as the follow-up was conducted: in fact several cancers were in advanced stage, thus with already thoracic and CNS involvement. Do the recurrence was considered a new lesion or a progression of an already present lesion?

In which manner the follow-up was conducted? Which type of exams were undertaken? CT scan, MRI, PET/CT other? Which timing was used to do radiological exams? 

Author Response

(The authors gave the same response as above.)

Round 2

Reviewer 3 Report

THE AUTHORS HAVE PARTIALLY ADDRESSED THE QUESTIONS OF REVIEWERS, THERE ARE SOME LIMITATIONS IN THE STUDY.

Author Response

There are no specific comments to respond to from this reviewer